# Contamination of Plant Foods with *Bacillus cereus* in a Province and Analysis of Its Traceability

**DOI:** 10.3390/microorganisms11112763

**Published:** 2023-11-14

**Authors:** Yingting Lin, Xiaoyan Cha, Charles Brennan, Jianxin Cao, Ying Shang

**Affiliations:** 1Faculty of Food Science and Engineering, Kunming University of Science and Technology, Kunming 650500, China; 20212114038@stu.kust.edu.cn (Y.L.); 20212114039@stu.kust.edu.cn (X.C.); charles.brennan@rmit.edu.au (C.B.); 2School of Science, Royal Melbourne Institute of Technology University, Melbourne 3000, Australia

**Keywords:** *Bacillus cereus*, food contamination, whole-genome sequencing, molecular traceability

## Abstract

*Bacillus cereus* is an important zoonotic foodborne conditional pathogen. It is found in vegetables, dairy products, rice, and other foods, thereby greatly endangering human health. Investigations on *B. cereus* contamination in China primarily focus on raw milk, dairy products, meat, and others, and limited research has been conducted on plant-based foodstuffs. The rapid development of sequencing technology and the application of bioinformatics-related techniques means that analysis based on whole-genome sequencing has become an important tool for the molecular-epidemiology investigation of *B. cereus*. In this study, we investigated the contamination of *B. cereus* in six types of commercially available plant foods from eight regions of a province. The molecular epidemiology of the isolated *B. cereus* was analyzed by whole-genome sequencing. We aimed to provide fundamental data for the surveillance and epidemiology analysis of *B. cereus* in food products in China. The rapid traceability system of *B. cereus* established in this study can provide a basis for rapid molecular epidemiology analysis of *B. cereus*, as well as for the prevention and surveillance of *B. cereus*. Moreover, it can also be expanded to monitoring and rapid tracing of more foodborne pathogens.

## 1. Introduction

*Bacillus cereus* (Frankland and Frankland 1887) (*Bacillales, Bacillaceae*) is a parthenogenic anaerobic Gram-positive bacterium. It is one of the *Bacillus* species, with a short rod-like circumference, flagella, and oval-shaped budding spores usually located centrally or somewhat offset [1]. Most of the *B. cereus* exhibit different forms depending on the environment in which they are observed. They can be isolated from environmental reservoirs, such as soil, marine sediment, seawater, and plants [2,3]. Some *B. cereus* strains have high virulence potential and are involved in various invasive and lethal infections [4]. *Bacillus cereus* causes two different types of gastrointestinal disorders, namely, vomiting and diarrhea syndrome, which are caused by different types of toxins [5]. Vomiting is caused by the vomiting toxin it produces, and diarrhea is caused by a hemolytic enterotoxin (hemolysin BL, Hbl), a non-hemolytic enterotoxin (Nhe), a cytotoxin K (CytK), and several other enterotoxins [5,6]. *Bacillus cereus* can grow rapidly under a wide range of conditions and can remain dormant under adverse environmental conditions. Thus, it can survive the heating of food and rapid multiply upon entering the body, leading to human infection [7].

*Bacillus cereus* is an important foodborne pathogen that is ubiquitous in nature and can be isolated from many different foods and food ingredients [8,9]. In 2018, the EU had 98 foodborne illness outbreaks, with *B. cereus* ranking fifth behind *Salmonella* (Salmon 1885), *Campylobacter* (Marshall et al. 1983), *norovirus* (Kapikian 1972), and *Staphylococcus aureus* (Ogston. 1882) [10]. A study that analyzed 3654 food samples tested in connection with the occurrence of foodborne illnesses detected the presence of *B. cereus* in 187 of these samples, with a contamination rate of 5% [11]. In addition to foodborne illnesses such as diarrhea and vomiting, *B. cereus* can cause various types of illnesses, including fulminant bacteremia, meningitis, pneumonia, gas gangrene infection, and endophthalmitis [12,13].

*Bacillus cereus* isolates have high genetic diversity. Conventional phenotypic identification methods distinguish between *B. cereus* and its constituent members [14]. Genome-based taxonomic criteria are increasingly being accepted for bacterial classification and species identification [15]. With the rapid spread of whole-genome sequencing technology, the whole-genome information of *B. cereus* has become widely available. Genome-wide data-based typing methods for *B. cereus* provide high resolution. These include Core Genome multilocus sequence typing (cgMLST), single-nucleotide polymorphism (SNP) analysis, and a genome-wide average nucleotide identity-setting threshold-based method [16,17,18]. Recent studies have found that for different taxonomic systems, *B. cereus* currently includes a total of 23 published subspecies [19].

The traceable typing of bacteria allows for the precise differentiation of bacteria, and certain differences exist in biological properties among different strains of the type within the same species, leading to various measures for their control. By studying the similarity of bacteria in an event, the type and source of the bacterial strain causing the infection can be determined, allowing for the most appropriate treatment and prevention protocols to be developed and thus prevent an outbreak. The earliest methods of bacterial trace typing include serotyping, phage typing, and high pulse-field gel electrophoresis (PFGE). However, such trace-typing methods are often accompanied by drawbacks such as cumbersome operation, low resolution, and weakly transferable results [20]. Whole-genome-sequencing (WGS)-based bacterial typing methods have great discriminatory power and data transferability for the epidemiological surveillance and clinical treatment of bacteria. With the rapid development of sequencing technologies, the cost of WGS decreases significantly, and WGS-based strain analysis methods are extensively used for bacterial biological characterization [21]. WGS has a higher resolution than PFGE in several strains in which horizontal transfer is known to exist. WGS-based bacterial-typing methods have begun to replace molecular methods such as PFGE as the “gold standard” for bacterial subtyping [22]. WGS has become an important tool for investigating foodborne disease outbreaks, and some countries have incorporated WGS into their national food safety control systems. In 2019, China established a WGS-based molecular traceability network for foodborne diseases, providing a strong guarantee for the prevention and control of foodborne pathogenic bacteria [23].

Further studies on strain WGS can provide insights into a strain’s resistance profile, virulence gene evaluation, and molecular typing characteristics. Comparative WGS analyses with routine diagnostic workups of tuberculosis mycobacteriosis have shown that WGS predicts species with 93% accuracy and offers drug sensitivity accuracy. WGS could reportedly diagnose a case of multidrug-resistant conjugate mycobacteriosis before the end of routine diagnostics, and WGS traceability showed that the analyzed samples were closely related to *Mycobacterium tuberculosis* (Zopf 1883) in the United Kingdom [24]. When PFGE and WGS are used to compare *Listeria monocytogenes* (E. Murray et al. 1926) isolates, WGS found that two *L. monocytogenes* strains with different geographic origins but closely related PFGE profiles significantly differed in terms of antibiotic and heavy metal resistance determinants, as well as removable genetic elements [25]. Molecular surveillance using WGS can be an important complement to the phenotypic surveillance of antibiotic resistance. WGS can also provide insights into the genetic basis of a strain’s resistance mechanisms to antibiotics, as well as into the evolution and population dynamics of pathogens on different spatial and temporal scales [26]. Thus, WGS has become an important tool for public health surveillance and molecular epidemiological studies of infectious diseases. It enables an accurate geographical description of transmission and the ability to monitor pathogen incidence at the genotype level. Coupled with epidemiological and environmental surveys, it can provide the ultimate solution for tracing the source of epidemic infections [27].

## 2. Materials and Methods

### 2.1. Materials

The experimental samples were obtained from farmers’ markets, supermarkets, and food factories in eight districts of a province. To understand the potential public health threat within the food supply chain, 273 samples were collected, including 6 major categories of plant foods: wild mushrooms (32), soybean products (76), fresh vegetables (80), pickled vegetables (47), cereals (4), and frozen products (34). The sampling status is shown in Table 1.

### 2.2. Isolation and Identification

All samples were isolated and identified for *B. cereus* with reference to GB 4789.14-2014 “National Safety Standard [28] for Food-Microbiological Examination of Food-Test for *Bacillus cereus*” for the collected samples.

### 2.3. Whole-Genome Sequencing and Genome Assembly

#### 2.3.1. DNA Library Construction

DNA libraries were constructed using QIAseq FX DNA Library Kits, a FastPure Gel DNA Extraction Mini Kit, and a KAPA Hyper Prep Kit for genomic DNA fragmentation, with fragmentation product cut-gel recovery, end repair, splice ligation, and magnetic bead purification, respectively. The products and libraries were quality controlled by 2% agarose gel electrophoresis. Concentration quantification of the fragmented DNA, initial quantification of the library, and quality control of the constructed library were performed using a calibrated Invitrogen Qubit 4.0 fluorescence quantification instrument. A redundancy of one or two cycles was applied to the calculated maximum number of amplification cycles to ensure a successful library amplification session and a high final concentration of the DNA library. Finally, the amplified library was purified using 1×scale magnetic beads to remove any residual primers, enzymes, connectors, and organic reagents from the library.

#### 2.3.2. On-Board Sequencing

The whole-genome sequencing of *B. cereus* isolates on board was conducted by Shanghai Tianhao Biotechnology Co. (Shanghai, China) Whole-genome de novo sequencing of PE150 was performed using an Illumina NovaSeq 6000 (Illumina Inc., San Diego, CA, USA) sequencer.

#### 2.3.3. Raw Data Processing

As the Illumina NovaSeq 6000 raw data contained some lower-quality data, quality clipping of the raw data was required for more accurate subsequent assemblies. Trimmomatic (https://github.com/timflutre/trimmomatic, accessed on 20 December 2022) was used to process some of the lower-quality *B. cereus* whole-genome data. FastQC (https://github.com/s-andrews/FastQC, accessed on 20 December 2022) was used to check the quality of the sequence information before and after filtering the sequence information for quality testing [29].

#### 2.3.4. Genome Assembly

De novo assembly of the *B. cereus* genome was performed using Shovill, a collection of programs containing the assembly tools needed for genomic transfer.

### 2.4. Genome-Wide Bioinformatics Analysis

#### 2.4.1. Subspecies Typing

*Bacillus cereus* genome information was first downloaded from the Genome Taxonomy Database (GTDB). Then, a local mash sketch classification database was constructed on the CentOS 8.3 server, and Mash (https://github.com/marbl/Mash, accessed on 20 December 2022) was used to quickly estimate global mutation distances between *B. cereus* genomes, as well as to perform genome clustering to obtain *B. cereus* subspecies classification information.

#### 2.4.2. MLST

Multilocus sequence typing (MLST) was performed using seven housekeeping genes (glp, gmk, ilv, pta, pur, pyc, and tpi) based on the typing scheme of *B. cereus* provided by the PubMLST database (https://pubmlst.org/, accessed on 20 December 2022). The information from the cgMLST database of *B. cereus* downloaded from the MLST database and the whole-genome information of 73 strains of *B. cereus* isolated in this study were used as the local *B. cereus* MLST database. Sequence alignment was performed using BLAST+ to obtain the seven housekeeping gene numbers of each isolate and the corresponding sequence type (ST). The MLST results were clustered using PHYLOVIZ 2.0, and the unweighted pair-group average method with arithmetic mean was used to construct a minimum generating tree for *B. cereus*.

#### 2.4.3. Toxicity Gene Analysis

The Virulence Factor Database (VFDB) is a comprehensive virulence factor (VF) database that allows for the rapid identification of bacterial VFs [30]. In this study, the whole-genome information of 73 *B. cereus* strains was compared with the reference genome in the VFDB database using BLAST+ to obtain strain VF-related information, and the results were classified and counted.

#### 2.4.4. cgSNP Phylogenetic Analysis

First, the WGS of *B. cereus* was functional gene annotated using Prokka (https://github.com/tseemann/prokka, accessed on 20 December 2022) to obtain the strain amino acid sequence type (GBK file). Second, the sequences in the GBK file were analyzed for Orthologs by Roary (http://sanger-pathogens.github.io/Roary/, accessed on 20 December 2022), and SNP results were obtained for 73 strains of *B. cereus*. Third, sequence alignment of core genomic SNP sites was performed using SNP-Sites (https://github.com/sanger-pathogens/snp-sites, accessed on 20 December 2022) to finalize the cgSNP phylogenetic tree.

In addition, to obtain accurate phylogenetic evolutionary relationships, a comparative analysis of the origin, subspecies identification, resistance genes, virulence genes, and MLST of 73 *B. cereus* strains was also carried out to construct a phylogenetic tree using IQTree (https://github.com/Cibiv/IQ-TREE, accessed on 20 December 2022).

#### 2.4.5. Establishment of a Molecular Traceability System for *B. cereus*

To achieve rapid molecular traceability of *B. cereus*, this study combined the published genome-wide information of 2900 *B. cereus* strains and constructed a rapid traceability workflow for *B. cereus* based on mutational distances among the sequences analyzed by Mash.

## 3. Results

### 3.1. Isolation and Identification of B. cereus

According to the color and morphology of the colonies, the suspected *B. cereus* colonies were selected from the *B. cereus* chromogenic medium (Central Bio-Engineering, Shanghai, China) for 16S rRNA sequencing, and the sequences of the strains were uploaded to NCBI for BLAST comparison. Results showed that among the 84 suspected *B. cereus* strains isolated from the samples, 73 had >99% homology to published *B. cereus* sequences in NCBI, and they were judged as *B. cereus*-positive. The remaining 11 suspected *B. cereus* strains had greater than 99% homology to other species in the database (6 *Bacillus proteus* (Hauser 1885), 3 *Lactococcus lactis* (Lister 1873), and 2 *Acinetobacter* (Brisou 1954)), and they were determined to be *B. cereus*-negative. Therefore, among the 273 samples collected, 56 were positive for *B. cereus*, and 73 strains were isolated.

### 3.2. Detection of B. cereus in Samples

#### 3.2.1. Results of *B. cereus* Detection in Different Regions

The contamination of *B. cereus* in plant foods in one province varied significantly by region. The positive samples of *B. cereus* in plant foods from different regions are shown in Table 2. The contamination rate of *B. cereus* in Region A was 42.86%, including 33.33% for pickled vegetables and 100% for frozen foods. One sample each was collected for soybean products and fresh vegetables, and no *B. cereus* contamination was detected. The contamination rate was 30.3%, including 66.67% for wild mushrooms, 20% for soybean products, 27.27% for fresh vegetables, 10% for pickled vegetables, and 75% for frozen products. A total of 12 samples were collected from Area C. All samples were wild mushrooms, and the contamination rate of positive samples was 41.67%. In Area D, 58 samples were collected, and 16 samples were contaminated with *B. cereus*, with a contamination rate of 27.59%. About 10% of the samples were contaminated with wild bacteria, 40% were contaminated with fresh vegetables, and 37.5%, 27.27%, and 21.43% were positive for soybean products, pickled vegetables, and frozen food, respectively. The detection rate of positive samples from Region E was low, with only 5 out of 49 samples collected with *B. cereus* contamination (10.2%), including 2 wild mushrooms (40%), 2 soybean products (14.29%), and 1 frozen food (25%), whereas no *B. cereus* contamination was found in fresh and preserved vegetables. Notably, 53 samples were collected from Region F, and none of them were found to be contaminated with *B. cereus*. Contamination was also not evident in Region G. Among the 42 samples collected, only 5 samples were contaminated, with a contamination rate of 11.9%, including 4.17% for soybean products and 25% for fresh and preserved vegetables. No contamination was found in frozen foods. A total of 12 samples were collected from Region H, with a contamination rate of 75%.

Within the eight areas studied, the highest rate of *B. cereus* contamination was found in Area H. This result can be attributed to the contaminated processing environment of the two food factories sampled. In order, the remaining areas include Areas A (42.86%), C (41.67%), B (30.30%), and D (27.59%). No *B. cereus* was found in the samples collected from Area F. The contamination of *B. cereus* in Areas G and E was also relatively low. Therefore, regions should pay attention to the need for sound food safety protection policies, and regulations in areas with high levels of contamination should be strengthened. Due to the limited number of samples collected in this study, the overall percentage of contamination in some areas may vary considerably. Further investigation of *B. cereus* contamination in such areas is needed.

#### 3.2.2. Detection of Different Types of *B. cereus*

When the prevalence of *B. cereus* was analyzed by food category and collection site, some variations in the prevalence of *B. cereus* in different food categories were found. Positive samples were detected in all food categories, as shown in Table 3. The *B. cereus* contamination rate for wild mushrooms was 31.25%, whereas the detection rates for supermarkets and farmers’ markets were 28.57% and 33.33%, respectively. A total of 76 samples of soybean products were collected, and 9 positive samples were found, including 3 in supermarkets (7.89%), 3 in farmers’ markets (9.09%), and 2 in food factories (40%). The positive detection rate for *B. cereus* in fresh vegetables was 15%, including 77 (12.99%) in supermarkets and 3 (66.67%) in farmers’ markets. The contamination rate of *B. cereus* in pickled vegetable samples was 23.40%, including 13.79% in supermarkets, 26.67% in farmers’ markets, and 100% in food factories. The contamination rate of frozen food was 29.41%, with 40.91% of the supermarket samples and 16.67% of the farmers’ market samples being contaminated. Four samples of cereals were collected, and all had *B. cereus* contamination, with a contamination rate of 100%. Analysis of the six food categories tested revealed that the cereal category had a 100% contamination rate.

Apart from cereal samples, wild bacteria had the highest contamination rate, followed by frozen foods, preserved vegetables, and fresh vegetables. The lowest level of *B. cereus* contamination was in soya products. Wild mushrooms were observed to be susceptible to infection with *B. cereus* in environmental soil, probably due to their special growth habit and generally prolonged contact with soil. The contamination rate of quick-freeze food was 29.41%. Spores are dormant at low temperatures and can grow rapidly once they are moved into a suitable environment, so food should be heated sufficiently when consuming this type of food. The contamination rate of *B. cereus* in pickled vegetables (23.4%) was higher than that in fresh vegetables (15%), which may be due to the complex processing of pickled vegetables and the fact that many pickled vegetables are pickled by farmers themselves. The processing of these vegetables may not meet hygienic conditions, and they are contaminated with *B. cereus* during processing. Among the several species tested, soybean products had the lowest rate of positive *B. cereus* samples at 11.84%. Few relevant studies have been conducted on the prevalence of *B. cereus* in soybean products. Thus, this study provides a reference for the prevention and control of *B. cereus* in soybean products.

#### 3.2.3. Detection of *B. cereus* in Different Sites

Differences existed in *B. cereus* among various sampling sites. The detection of *B. cereus* in different sampling sites in the eight regions is shown in Table 4, and the proportion of contaminated *B. cereus* in supermarkets, farmers’ markets, and food factories is shown in Figure 1. A total of 180 samples were collected from supermarkets in each region, and 30 samples were positive for *B. cereus*, with a contamination rate of 16.67%. Among them, Area A had the most serious contamination, with a contamination rate of 42.86%, followed by Areas B (33.33%), D (30.77%), and G (21.05%). A lower level of contamination was found in Area E supermarkets, with a contamination rate of 10.20%. The contamination of *B. cereus* in farmers’ markets was more serious than that in supermarkets, with a contamination rate of 20.99%. Region C had a more serious contamination, with a positive sample rate of 41.67%. Region B and D had the same level of contamination at 25%, and region G had a lower level of contamination at 4.35%. No *B. cereus* was found in supermarkets or farmers’ markets in Area F. The two food factories in Area H were a pickle factory and a rice noodle factory, with a contamination rate of 75%. Eight samples were collected from the pickle factory, wherein five samples were positive for *B. cereus* (a contamination rate of 62.5%). Four samples were collected from the rice noodle factory, and *B. cereus* was detected in all samples.

The level of *B. cereus* contamination in supermarket samples was lower than that in farmers’ markets. The highest level of *B. cereus* contamination was found in Area H, where 75% of the samples were positive for *B. cereus*. This finding may also be related to the fact that the two food factories sampled were small factories with poor overall hygiene conditions, no strict sterilization system, and poor awareness of food safety prevention and control among the staff involved. These factors may have led to the environmental contamination and overall high detection levels. Eight samples were collected from the pickle factory, and the detection rate of positive samples was 62.5%, which was a high-contamination situation. The detection of *B. cereus* may also be related to the curing time. In the detection of *B. cereus* in curd products with different shelf-lives, we found that when the samples were infected with *B. cereus* at the time of production, the high resistance of *B. cereus* caused it to grow slowly in the first few months under the conditions of high salt and low oxygen. However, the number of *B. cereus* was gradually reduced again after 6 months, when the oxygen was basically depleted [31]. In the sampled rice noodle factory, the contamination of *B. cereus* reached 100%, and the samples involved all stages of production from raw materials to finished products, a situation that can be caused by the presence of *B. cereus* contamination already within the factory. Some researchers have similarly found high levels of *B. cereus* contamination in rice noodle products such as rice noodles and liangpi [32]. Therefore, paying attention to such foods in daily dietary consumption is important. They should be fully heated before consumption to prevent food poisoning.

### 3.3. Statistics and Assembly of Sequencing Results

The quality-controlled sequences were subjected to *B. cereus* genome assembly, and the results for 73 *B. cereus* isolates are shown in Figure 2. The GC% of all *B. cereus* isolates was around 35%, in line with the GC ratio of *B. cereus*. The size of the spliced genome ranged within approximately 5–6 M, consistent with the genome size of *B. cereus* published on NCBI and GTDB. The number of contigs ≥ 25,000 bp after splicing was above 20, most N50s were above 100 000, and the L50 values were small. Therefore, the genome assembly results of the 73 *B. cereus* isolates were good and can be used for subsequent experimental analysis.

### 3.4. Molecular Typing of B. cereus

#### 3.4.1. Results of *B. cereus* Subspecies Typing

The subspecies identification of 73 *B. cereus* isolates was performed by Mash. Seven *B. cereus* subspecies were identified, namely, *Bacillus albus* (Travers 1987) (3 strains), *Bacillus anthracis* (Koch 1876) (2 strains), *B. cereus* (23 strains), *Bacillus paranthracis* (Liu et al. 2017) (40 strains), *Bacillus thuringiensis* (Berliner 1915) (2 strains), *Bacillus toyonensis* (Jiménez 2013) (2 strains), and *Bacillus tropicus* (Liu et al. 2017) (1 strain). The subspecies typing of 73 *B. cereus* isolates further revealed that the *B. cereus* isolation method using a combination of *B. cereus* selective media and 16S rRNA was a highly suitable means of *B. cereus* isolation and identification.

#### 3.4.2. MLST Results

The MLST method based on housekeeping genes allows for the rapid inference of population structures and evolutionary relationships between species. It is a common method of studying the epidemiological situation of bacteria [33]. The MLST clustering results for the 73 strains of *B. cereus* in this study are shown in Table 5. The 73 samples were divided into 34 ST types, among which ST26 (11 strains) and ST1431 (8 strains) were the major ones. Four clonal complexes (CCs) were included: CC142 (11 strains), CC111 (1 strain), CC205 (9 strains), and CC97 (1 strain). The 73 *B. cereus* MLST results showed that *B. cereus* was extensively distributed in all regions of a province and in all types of plant foods as a whole, and no obvious clustering was found among *B. cereus* from the same region or food category. The main ST types were detected in all regions, probably due to the trade of related foods and the frequent movement of people, which led to the spread of clonal strains. Several studies have shown that the clustering of *B. cereus* in various regions is not obvious, and the distribution of ST types is wide-ranging [34,35].

#### 3.4.3. MLST Results of *B. cereus* in Different Regions

Using PHYLOVIZ 2.0, a minimal spanning tree was constructed for the MLST results of *B. cereus* from different regions, as shown in Figure 3. The distribution of ST types of strains from different regions was relatively scattered, and no obvious clustering was found. More strains had the same ST type in different regions, among which ST26 and ST142 contained more strains and were primarily distributed throughout regions A, B, C, D, and H. ST177 was distributed in regions A, B, and D, indicating that clonal transmission of *B. cereus* may have occurred in different regions.

#### 3.4.4. MLST Results of *B. cereus* in Different Foodstuffs

The MLST results of 73 strains of *B. cereus* compared with the food groups are shown in Figure 4. Each food group corresponded with various ST types. Frozen foods contained eight ST types, among which ST177 (4 strains) was the main one. Pickled vegetables had 11 ST types, and wild mushrooms had 8 ST types, with no significant ST types present in either food. Soybean products contained 10 ST types, and cereals contained 4 ST types, with only 1 strain of each ST type. Fresh vegetables contained 10 ST types, with ST26 being the predominant ST type (5 strains). ST1431 was present in all five food groups.

### 3.5. Toxicity Gene Detection Results

The carriage of the major virulence genes of *B. cereus* was analyzed by whole-genome sequencing. We selected 14 virulence genes strongly associated with the pathogenic characteristics of *B. cereus* as follows: the vomitoxin genes *cesA*, *cesB*, *cesC*, *cesD*, *cesH*, *cesP*, and *cesT*; the hemolytic enterotoxin genes *HblA*, *HblC*, and *HblD*; the non-hemolytic enterotoxin genes *NheA*, *NheB*, and *NheC*; and the cytotoxin gene *CytK*. These were analyzed for carriage in 73 strains of *B. cereus*, and the results are shown in Table 6. Differences existed in the detection of the virulence genes. Non-hemolytic enterotoxin-like genes were detected at the highest rate, with all strains carrying *NheB* and *NheC*. The gene *NheA* was also detected at a rate of 98.63%, followed by *CytK*, with a detection rate of 63.01%. An additional 34 strains of *B. cereus* carried *HblA*, *HblC*, and *HblD*, with no separate cases detected. The genes encoding vomiting toxins were the least detected, with *cesH* detected at a rate of 32.88% among these seven genes and the rest at around 21%. The overall detection of virulence genes was in accordance with that reported by the World Health Organization.

### 3.6. cgSNP Phylogenetic Analysis

cgSNP typing is a high-resolution, reproducible method for genotyping and strain-relatedness analysis. Together with the disseminability of its data, it is used in many countries for the epidemiology investigations and traceability analysis of foodborne microorganisms [36,37]. The amino acid sequences of 73 *B. cereus* strains were analyzed using Roary, and 391 SNP loci were detected. cgSNP phylogenetic evolutionary trees were obtained by sequence alignment based on these loci, and IQtree was used to summarize other relevant information from the samples to generate a *B. cereus* phylogenetic tree. Results are shown in Figure 5. The 73 *B. cereus* strains were divided into seven main taxa (Groups), with the exception of Groups 1, 2, and 3, which were independent branches. The remaining taxa were divided into different subdivisions. The phylogenetic tree clustering showed that strains of the same ST type were closely related to each other, for example, ST26 and ST1431 were clustered in one branch. The expression levels of virulence genes may be correlated with the clustering results, e.g., Group 7 strains all contained the *Ces* gene encoding vomitoxin, whereas strains bc-177/bc-142/bc-143/and bc-100/bc-245/bc173 in the same branch were expressed at the same level. The clustering results of the SNPs and information on the origin of the strains showed that the branches did not show differences in regions and food groups. Results of the typing and clustering of strain GTDB subspecies showed that the cgSNPs of strains with the same subspecies were in the same cluster except for some strains (bc-51 and bc-243). No specific correlation was found between the expression levels of drug resistance genes and SNP clustering results.

The 73 strains of *B. cereus* had rich genetic diversity. Phylogenetic analysis of *B. cereus* by different analytical methods revealed that the cgSNP clustering results of the strains were correlated with virulence gene expression levels, ST type, and GTDB subspecies classification. This finding may be related to genotypic sequence differences. No correlation was found between the results of cgSNP clustering analysis and strain origin, suggesting that *B. cereus* was widely spread within a province, and no obvious geographical differences existed among strains. Moreover, extensive horizontal gene transfer occurred among them.

### 3.7. Establishment of a Traceability System for B. cereus

To achieve rapid traceability of *B. cereus*, this study established a set of *B. cereus* rapid traceability analysis flow, which can achieve the rapid epidemiological investigation of *B. cereus*. The workflow included a series of data pre-processing stages, such as filtering, assembly, splicing, and decontamination of Illumina offline raw data. By integrating the characteristic gene information of *B. cereus* from public databases such as NCBI, GTDB, VFDB, and Comprehensive Antibiotic Resistance Database, an automated analysis module of *B. cereus* drug resistance genes, virulence factors, and metabolic pathway analysis was established. The module enabled in-depth research on the information of the disease-causing potential and drug resistance mechanism of *B. cereus*. The database currently contains the whole-genome information of 2900 published *B. cereus* strains. Combined with cgMLST, cgSNP, and other typing methods, it can realize the rapid molecular traceability analysis of *B. cereus*. The workflow comprises two work modules, Bact-typing and Panphylo, the relevant contents of which are shown in Figure 6. When using the workflow, the user first needs to formulate the raw data information, primarily including the following three files: mapping_file.txt, metadata.txt, and raw_data. After configuration, the terminal can be activated to complete the relevant analysis according to the actual needs. The whole workflow can be completed within only 5 h. The user can select the corresponding method for fast and scientific analysis according to the actual needs, thereby providing a strong guarantee for the prevention and monitoring of *B. cereus*.

## 4. Discussion

Six types of commercially available plant foods were sampled in eight regions of a province, and 273 food samples were obtained. A total of 73 strains of *B. cereus* were isolated, purified, and identified. They were subjected to molecular epidemiology studies through Illumina II whole-genome sequencing, and molecular epidemiology studies were conducted on the whole-genome sequences of the 73 strains of *B. cereus* by bioinformatics methods. The overall situation of *B. cereus* contamination in plant foods in various regions of a province is complex, with obvious differences between regions and establishments, and attention should be paid to improving the food safety protection system and strengthening the supervision, prevention, and control of seriously contaminated regions and establishments. There are many reasons for the differences, which may be due to the growth habit of plant foods, the production process, the environment of production and operation, as well as the growth characteristics of *B. cereus* itself.

The *B. cereus* population includes many closely related species, with pathogenic harmful bacteria as well as probiotic bacteria with positive effects on human health and agricultural development, so the precise differentiation of *B. cereus* subspecies is essential for public health risk assessment as well as for industrial and agricultural development. *Bacillus paranthracis* is a new, recently delineated subtype that has been found in various environments [38,39]. Some researchers have identified five peroxidase genes that can promote the growth of lactic acid bacteria during fermentation in the genome of *B. paranthracis* isolated from fermented rice bran [40]. In the present study, 13 strains of *B. paranthracis* were also detected from pickled vegetables, and these strains may have similarly positive effects on the fermentation of vegetables.

As an important foodborne pathogen, *B. cereus* can cause various degrees of food poisoning. Notably, we detected *B. thuringiensis*, which is widely used for agricultural pest control, in the presence of virulence genes. *Bacillus thuringiensis* isolated from fermented soybeans was found to contain diarrhea-type enterotoxin genes, consistent with our results [41]. This finding suggested that *B. thuringiensis* had the same pathogenic potential, and relevant industries should be aware of the presence of such subspecies when using or regulating them to prevent the occurrence of diseases.

In this study, epidemiological investigations and traceability analyses of *B. cereus* using MLST typing and cgSNP typing showed that the 73 strains of *B. cereus* as a whole were widely distributed, with rich genetic diversity, and yet no correlation was found with the source of the strains. However, some researchers have analyzed 44 whole-genome sequences of *B. cereus* isolates from different sources in Japan and found that *B. cereus* from different regional sources are also closely related genetically [42]. Finally, the prevalence of *B. cereus* should be continuously monitored to achieve rapid epidemiological analyses of *B. cereus* and effective prevention of foodborne illnesses, providing a strong guarantee for the surveillance of *B. cereus*.

## 5. Conclusions

In this study, the prevalence of *B. cereus* in plant foods was investigated, and the presence of other species of bacterial contamination of the samples was found during testing. Therefore, it is necessary to identify the microbial communities of the samples, analyze the presence of other pathogenic bacteria, and study the population abundance and potential risk of pathogenicity of the samples. Due to the limited number of samples collected in this study, in order to obtain an accurate contamination rate of *B. cereus* in plant foods, the sampling volume may be increased appropriately for further analyses. In addition, the genomic characteristics of *B. cereus* subspecies are extremely similar to each other, and the GTDB typing scheme is based on multiphase genomics with gene-averaged nucleotide identity for the standardized typing scheme to classify and name the subspecies of *B. cereus*, which may lead to the similarity or concordance of physiological characteristics between different subspecies, thus affecting the judgement of the functional characteristics of the strains. When the virulence genes were analyzed, it was found that the characteristics of virulence genes were not obvious among different subspecies, so the physiological characteristics of the strains can be studied subsequently for different subspecies of *B. cereus*, which can provide a scientific basis for the classification of *B. cereus*. In order to ensure the effectiveness of traceability and achieve rapid traceability analysis of *B. cereus*, it is necessary to continuously update the content of the traceability database of *B. cereus* established in this study and the related bioinformatics analysis software (Version 1.0) in the server, so as to achieve the purpose of accurate traceability of *B. cereus*.

## Figures and Tables

**Figure 1 microorganisms-11-02763-f001:**
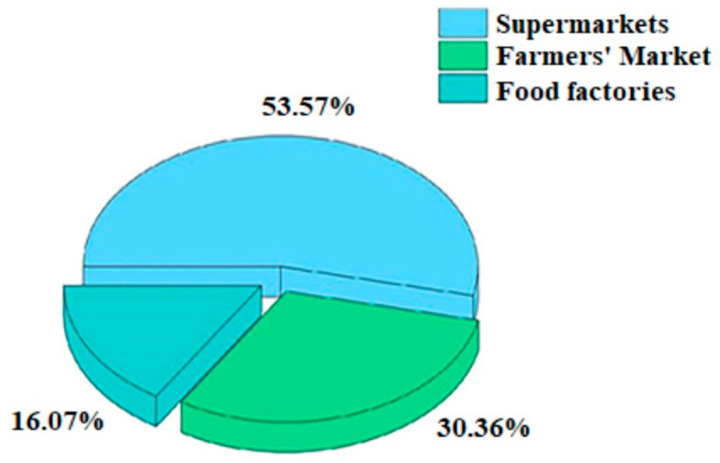
Proportion of *B. cereus* contamination in different places.

**Figure 2 microorganisms-11-02763-f002:**
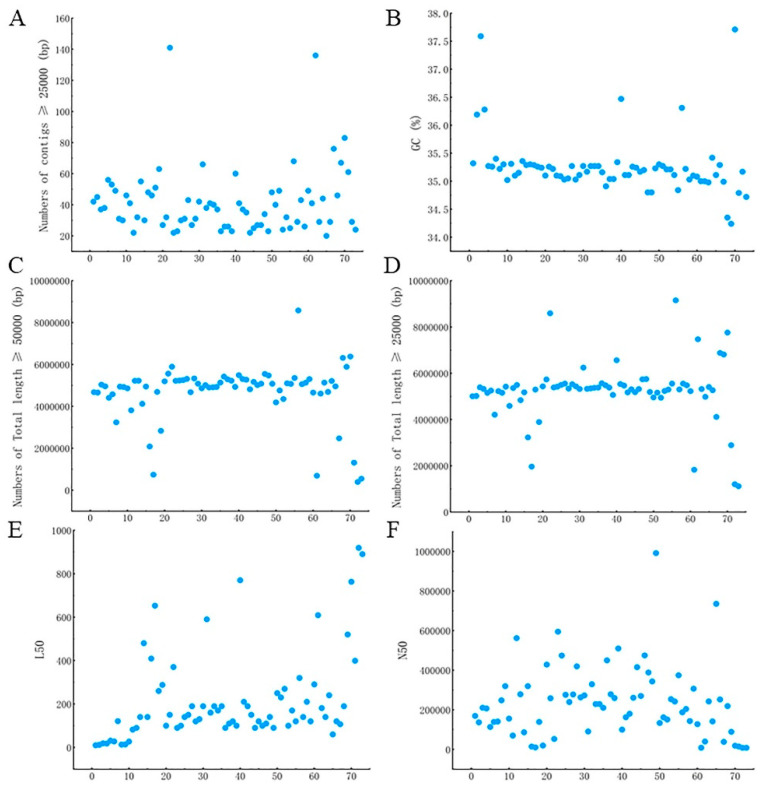
Whole-genome assembly information of 73 *B. cereus* isolates. (**A**) Number of contigs greater than 25,000 bp after assembly; (**B**) genomic GC content; (**C**) total length of contigs greater than 50,000 bp after assembly; (**D**) total length of contigs greater than 25,000 bp after assembly; (**E**) L50 value after assembly; (**F**) N50 value after assembly.

**Figure 3 microorganisms-11-02763-f003:**
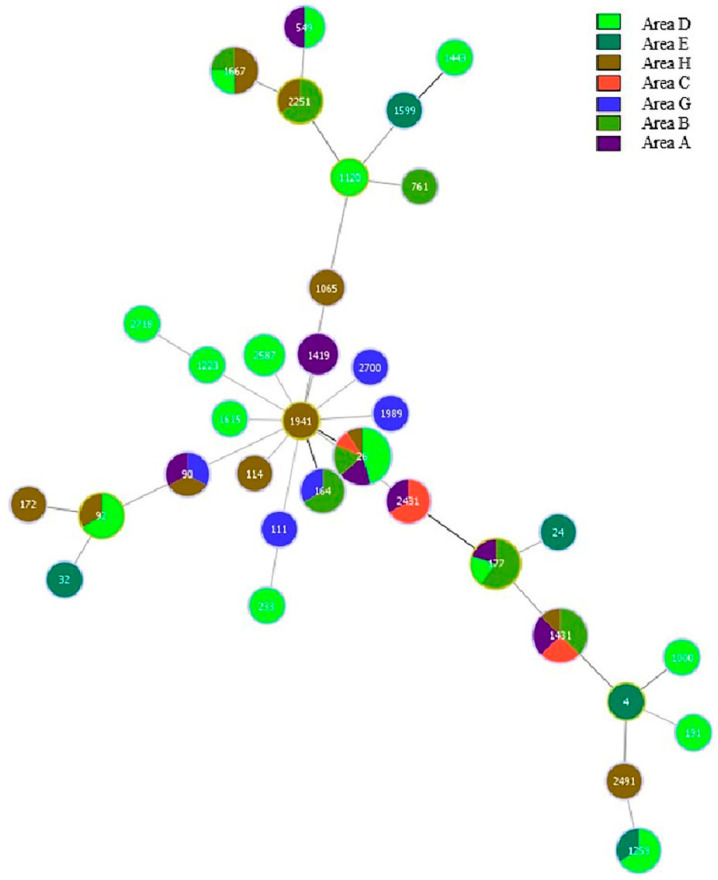
MLST cluster analysis of *B. cereus* in different regions.

**Figure 4 microorganisms-11-02763-f004:**
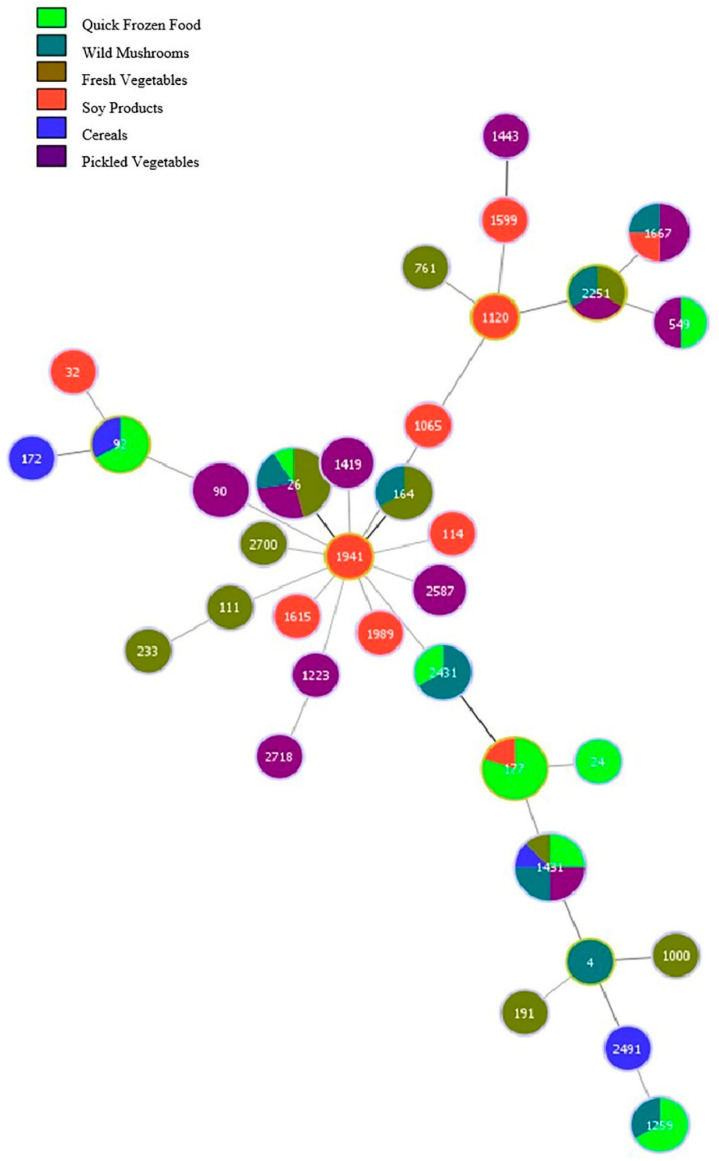
MLST Cluster analysis of *B. cereus* in different foods.

**Figure 5 microorganisms-11-02763-f005:**
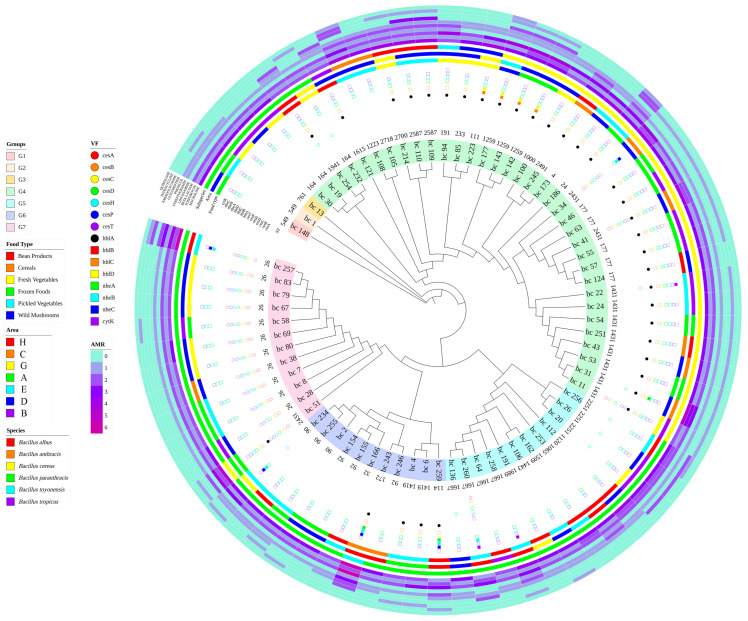
cgSNP clustering analysis of 73 *B. cereus*.

**Figure 6 microorganisms-11-02763-f006:**
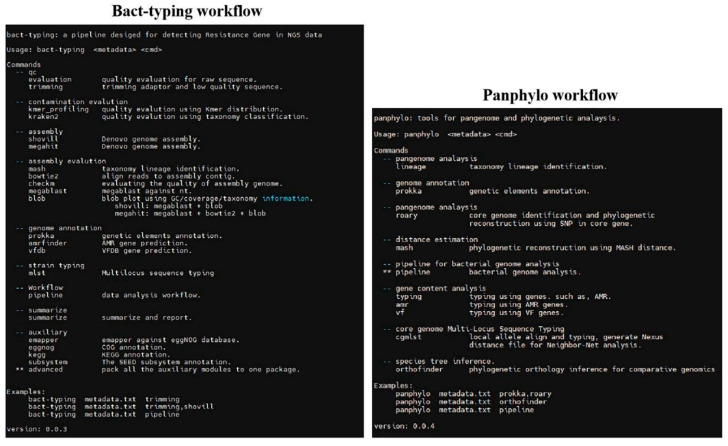
Traceability analysis workflow for *B. cereus*.

**Table 1 microorganisms-11-02763-t001:** Summary of food sampling (unit: piece).

Samples	Wild Mushrooms	Soybean Products	Fresh Vegetables	Pickled Vegetables	Quick-Freeze Food	Cereals	Total
Area A	—	1	1	9	3	—	14
Area B	3	5	11	10	4	—	33
Area C	12	—	—	—	—	—	12
Area D	10	8	15	11	14	—	58
Area E	5	14	22	4	4	—	49
Area F	2	19	19	6	7	—	53
Area G	—	24	12	4	2	—	42
Area H	—	5	—	3	—	4	12
Total	32	76	80	47	34	4	273

**Table 2 microorganisms-11-02763-t002:** Prevalence of *B. cereus* in various regions (unit: piece).

Samples	Wild Mushrooms	Soybean Products	Fresh Vegetables	Pickled Vegetables	Quick-Freeze Food	Cereals	Total	Pollution Rate (%)
Area A	—	1	0	3	3	—	6/14	42.86
Area B	2	5	3	1	3	—	10/33	30.30
Area C	5	—	—	—	—	—	5/12	41.67
Area D	1	3	6	3	3	—	16//58	27.59
Area E	2	2	0	0	1	—	5/49	10.20
Area F	0	0	0	0	0	—	0/53	0.00
Area G	—	1	3	1	0	—	5/42	11.90
Area H	—	2	—	3	—	4	9/12	75.00
Total	10/32	9/76	12/80	11/47	10/34	4/4	56/273	20.51

**Table 3 microorganisms-11-02763-t003:** Prevalence of *B. cereus* in different food types.

Samples	Supermarkets	Farmers’ Markets	Food Factories	Total	Pollution Rate (%)
Wild mushrooms	4	6	—	10/32	31.25
Soybean products	3	3	2	9/76	11.84
Fresh vegetables	10	2	—	12/80	15.00
Pickled vegetables	4	4	3	11/47	23.40
Quick-freeze food	9	2	—	10/34	29.41
Cereals	—	—	4	4/4	100.00
Total	30/180	17/81	9/12	56/273	20.51

**Table 4 microorganisms-11-02763-t004:** Prevalence of *B. cereus* in different sites (unit: piece).

Samples	Supermarkets	Farmers’ Markets	Food Factories	Total	Pollution Rate (%)
Area A	6	—	—	6/14	42.59
Area B	7	3	—	10/33	30.30
Area C	—	5	—	5/12	41.67
Area D	8	8	—	16/58	27.59
Area E	5	—	—	5/49	10.20
Area F	0	0	—	0/53	0.00
Area G	4	1	—	5/42	11.90
Area H	—	—	9	9/12	75.00
Total	30/180	17/81	9/12	56/273	20.51

**Table 5 microorganisms-11-02763-t005:** MLST results of 73 *B. cereus* strains.

Type ST	*glp*	*gmk*	*ilv*	*pta*	*pur*	*pyc*	*tpi*	Number (Plants)	Clonal_Complex
4	13	8	8	11	11	12	7	1	ST-142 complex
24	12	8	9	14	11	12	10	1	—
26	3	2	31	5	16	3	4	11	—
32	5	4	3	4	15	6	16	1	—
90	6	4	41	5	43	46	3	3	—
92	6	4	42	4	16	6	3	3	—
111	43	26	35	42	39	41	30	1	ST-111 complex
114	8	10	105	36	17	70	11	1	—
164	3	2	63	5	36	3	4	3	—
172	6	4	3	63	16	6	3	1	—
177	13	47	9	11	68	12	10	5	—
191	15	6	29	8	4	7	7	1	ST-97 complex
233	87	26	91	90	91	75	30	1	—
549	3	2	59	17	19	126	55	2	—
761	19	2	59	65	19	3	55	1	ST-205 complex
1000	13	8	8	58	122	12	7	1	ST-142 complex
1065	3	2	31	348	49	3	2	1	ST-205 complex
1120	19	2	31	17	19	3	2	1	ST-205 complex
1223	55	1	83	1	230	37	43	1	—
1259	14	8	9	11	9	88	8	3	—
1419	94	2	154	5	32	3	114	2	—
1431	13	8	8	11	9	12	10	8	ST-142 complex
1443	19	2	234	5	208	18	2	1	ST-205 complex
1599	19	2	234	5	19	18	2	1	ST-205 complex
1615	48	30	33	37	44	31	51	1	—
1667	53	2	59	5	47	3	216	4	—
1941	3	2	31	5	36	3	4	1	—
1989	324	2	122	5	19	3	91	1	ST-205 complex
2251	53	2	59	17	19	3	2	3	ST-205 complex
2431	13	47	9	11	68	3	10	3	—
2491	13	8	8	11	122	12	8	1	ST-142 complex
2587	239	142	269	180	237	149	37	2	—
2700	226	31	231	43	45	53	159	1	—
2718	61	38	1	1	18	37	19	1	—

**Table 6 microorganisms-11-02763-t006:** Carrying status of virulence genes in 73 *B. cereus* isolates.

Genotypes	Toxic Genes	Number of Carrier Strains (Strains)	Detection Rates (%)
Vomitoxins	*cesA*	16	21.92
	*cesB*	15	20.55
	*cesC*	16	21.92
	*cesD*	16	21.92
	*cesH*	24	32.88
	*cesP*	16	21.92
	*cesT*	16	21.92
Hemolytic enterotoxins	*HblA*	34	46.58
	*HblC*	34	46.58
	*HblD*	34	46.58
Non-hemolytic enterotoxins	*NheA*	72	98.63
	*NheB*	73	100
	*NheC*	73	100
Cytotoxins	*CytK*	46	63.01

## Data Availability

Data are contained within the article.

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
