# Peer review of "Contamination of Plant Foods with Bacillus cereus in a Province and Analysis of Its Traceability"

_microorganisms, 2023, doi:10.3390/microorganisms11112763_

Round 1

Reviewer 1 Report

Comments and Suggestions for Authors

The article shows originality and it is easy to read. the research was based on the isolation of B.cereus strains from multiple sources of feed and areas so as to reveal a possible traceability marker, but as it was shown no obvious differences were noticed among the geographical areas studied. The research was meticulous and the statistics used appropriate for the research. It deserves to be published. 

Author Response

Thank you very much for your recognition of our work and its significance.

Reviewer 2 Report

Comments and Suggestions for Authors

The manuscript by Chinese and Australian colleagues deals with the identification of important foodborne pathogen, bacterium Bacillus cereusis in food products of plant origin. The presence of Bacillus cereus in food causes serious negative effects on human health. This is why the authors’ research is so important. The article is adequately illustrated; its theme is relevant to Microorganisms.

However, I have a few comments on this manuscript.

The main remark concerns the clarification of study areas. Where did the research take place? What does mean non-specific “eight districts of a province”? In sections 2.1. and 3.2.1. (in Tables too) it is necessary to indicate the localities where the samples were obtained. This is especially true for work with genetic analysis, which involves indicating geographic coordinates. So far it is only clear that the study was conducted somewhere in China.

The authors obtained rather voluminous qualitative results, while the discussion is poorly presented. The discussion needs to be expanded. Moreover, lines 416-422 are essentially results. Have similar studies been carried out in other countries? Make a comparison.

It is also necessary to highlight the Conclusion in a separate section.

According to the International Code of Zoological Nomenclature, at the first mention of all animal or parasite species, its full Latin name with the author and year of description should be given. For example, Listeria monocytogenes (E. Murray et al. 1926), Mycobacterium tuberculosis (Zopf 1883), in lines 299-301, etc.

For the main “hero” of the article, taxa of higher order should be given at the first mention. Bacillus cereus Frankland & Frankland 1887 (Bacillales, Bacillaceae)

All Latin names of species and genera of living organisms are always written in italics. For example: Bacillus species (line 26), Salmonella, Campylobacter, Staphylococcus aureus (lines 41,42), Mycobacterium tuberculosis (Zopf 1883) (line 80), lines 81, 82, 177, 212, 353-355, etc.

A sentence cannot begin with an abbreviated word (lines 31, 35, 39, 47, 134, etc). Therefore, the Latin name (Bacillus cereus) is given in full here.

The manuscript can be published in Microorganisms, but some corrections are needed.

Author Response

Response to Reviewer 3' Comments

1: Where did the research take place? What does mean non-specific “eight districts of a province”? In sections 2.1. and 3.2.1. (in Tables too) it is necessary to indicate the localities where the samples were obtained. This is especially true for work with genetic analysis, which involves indicating geographic coordinates.

Answer: Thank you for your valuable suggestion. We understand your suggestion, and your opinion is that specifying the region can make this paper more accurate. However, we have some concerns about disclosing the sampling location. Due to the particularity of the funded project and research content, it is not convenient to disclose the detailed sampling location, and then we choose to use the terms of a province and a region. But all the bacteria we obtained from this research are already stored in our culture bank, and if researchers need to use certain strains, we can provide them.

2: The authors obtained rather voluminous qualitative results, while the discussion is poorly presented. The discussion needs to be expanded. Moreover, lines 416-422 are essentially results. Have similar studies been carried out in other countries? Make a comparison.

Answer: Thank you for your advice. We have made corresponding modifications and adjustments to the discussion section as suggested (lines 404-438).

3: It is also necessary to highlight the Conclusion in a separate section.

Answer: Thank you for your suggestion. We have added a separate section of “Conclusion” (lines 439-457).

4: According to the International Code of Zoological Nomenclature, at the first mention of all animal or parasite species, its full Latin name with the author and year of description should be given. For example, Listeria monocytogenes (E. Murray et al. 1926), Mycobacterium tuberculosis (Zopf 1883), in lines 299-301, etc.

Answer: Thanks for your advice, we have given the full Latin name of the animal or parasite species first mentioned in the original manuscript, along with the author and the year of description (Line 41,42, 81-83, 178, 179, 300-303).

5: For the main “hero” of the article, taxa of higher order should be given at the first mention. Bacillus cereus Frankland & Frankland 1887 (Bacillales, Bacillaceae)

Answer: Thank you for your comment. We have revised as your suggestion (line 26).

6: All Latin names of species and genera of living organisms are always written in italics. For example: Bacillus species (line 26), Salmonella, Campylobacter, Staphylococcus aureus (lines 41,42), Mycobacterium tuberculosis (Zopf 1883) (line 80), lines 81, 82, 177, 212, 353-355, etc.

Answer: We have revised as your suggestion. (Line 27, 41, 42, 81-83, 178, 179, 213)

7: A sentence cannot begin with an abbreviated word (lines 31, 35, 39, 47, 134, etc). Therefore, the Latin name (Bacillus cereus) is given in full here.

Answer: Thanks for your advice. We have revised as your suggestion. (Line 31, 36, 39, 48, 135)
